# Harmonized in situ datasets for agricultural land use mapping and monitoring in tropical countries

Audrey Jolivot[1,2], Valentine Lebourgeois[1,2], Louise Leroux[3,4,5], Mael Ameline[1,2], Valérie Andriamanga[1,6], Beatriz Bellón[1,2], Mathieu Castets[1,2], Arthur Crespin-Boucaud[1,2], Pierre Defourny[7], Santiana Diaz[1,2], Mohamadou Dieye[8], Stéphane Dupuy[1,2], Rodrigo Ferraz[9], Raffaele Gaetano[1,2], Marie Gely[1,2], Camille Jahel[1,2], Bertin Kabore[10], Camille Lelong[1,2], Guerric le Maire[11,12], Danny Lo Seen[1,2], Martha Muthoni[13], Babacar Ndao[5,14], Terry Newby[15], Cecília Lira Melo de Oliveira Santos[16], Eloise Rasoamalala[1,6], Margareth Simoes[9], Ibrahima Thiaw[14], Alice Timmermans[7], Annelise Tran[1,2], Agnès Bégué[1,2]

[1]CIRAD, UMR TETIS, F-34398 Montpellier, France.
[2]TETIS, Univ Montpellier, AgroParisTech, CIRAD, CNRS, INRAE, Montpellier, France.
[3]CIRAD, UPR AIDA, Dakar, Senegal.
[4]AIDA, Univ Montpellier, CIRAD, Montpellier, France.
[5]Centre de Suivi Ecologique (CSE), Dakar, Senegal.
[6]Centre National de la Recherche Appliquée au Développement Rural (FOFIFA), Antsirabe, Madagascar.
[7]Université Catholique de Louvain (UCLouvain), Louvain-la-Neuve, Belgique.
[8]Institut Sénégalais de Recherches Agricoles (ISRA), Dakar, Senegal.
[9]Brazilian Agricultural Research Corporation (EMBRAPA), Rio de Janeiro, Brazil.
[10]Independent consultant, Ouagadougou, Burkina Faso
[11]CIRAD, UMR Eco&Sols, F-34398 Montpellier, France.
[12]Eco&Sols, Univ Montpellier, CIRAD, INRAE, Institut Agro, IRD, Montpellier, France.
[13]Independent consultant, Nairobi, Kenya.
[14]Université Cheikh Anta DIOP de Dakar (UCAD), Dakar, Senegal
[15]Agricultural Research Council (ARC), Pretoria, South Africa.
[16]Interdisciplinary Center on Energy Planning, NIPE, University of Campinas, UNICAMP, Campinas, Sao Paulo 13083-896, Brazil.

*Correspondence to*: Audrey Jolivot (audrey.jolivot@cirad.fr)

**Abstract.** The availability of crop type reference datasets for satellite image classification is very limited for complex agricultural systems as observed in developing and emerging countries. Indeed, agricultural land use is very dynamic, agricultural census are often poorly georeferenced, and crop types are difficult to photointerpret directly from satellite imagery. In this paper, we present a database made of 24 datasets collected in a standardized manner over nine sites within the framework of the international JECAM (Joint Experiment for Crop Assessment and Monitoring) initiative; the sites were spread over seven countries of the tropical belt, and the number of data collection years depended on the site (from 1 to 7 years between 2013 and 2020). These quality-controlled datasets are distinguished by in situ data collected at the field scale by local experts, with precise geographic coordinates, and following a common protocol. Altogether, the datasets completed 27 074 polygons (20 257 crops and 6 817 noncrops, ranging from 748 plots in 2013 (one site visited) to 5 515 in 2015 (six sites visited)) documented by detailed keywords. These datasets can be used to produce and validate agricultural land use maps in the tropics. They can also be used to assess the performances and robustness of classification methods of cropland and crop

types/practices in a large range of tropical farming systems. The dataset is available at https://doi.org/10.18167/DVN1/P7OLAP (Jolivot et al., 2021).

## 1. Introduction

Land use and land cover (LULC), and their changes, are key information for studying and monitoring carbon and water cycles, and threats to biodiversity, and for establishing land-use planning and public policies. In particular, accurate mapping of
45 cropland and associated cropping practices is of primary importance for food security, agricultural and environmental monitoring and land management. However, cropland and crop-type mapping using Earth observation data is still challenging as it requires large sets of training and validation data, and as the land use (field limits and content) generally changes annually, even seasonally. Large datasets on cropping practices are available in the Global North, mainly thanks to agricultural policies that support annual census and provide tools for the digitization at field level using Very High Resolution remote sensing
imagery (e.g. the Land Parcel Identification System designed to implement common agricultural policy in the European Union, or the Cropland Data Layer of the National Agricultural Statistic Services of the United States Department of Agriculture). Such data sets provide a very large number of annotated surface samples reporting yearly crop types, which can often easily be integrated in reference data sets for land cover mapping systems at the cost of a relatively simple "cleansing and harmonization" procedure (Inglada et al., 2017). Despite the fact that the declarative nature of such annotations makes them
error-prone, such "noise" is typically compensated by the large number of available crop type samples. As arguable, no such large scale data base currently exists in most of the developing and emerging countries. Matter of facts, in these countries, cropland and crop types can be particularly difficult to map (Waldner et al., 2015) because the fields are often small to medium size (Fritz et al., 2015), because crops are easily confused with natural vegetation and fallows and because cropping systems are typically highly variable in time and space. Each farming system has its own specificities in terms of crop type and
composition, field size, cropping calendar, irrigated/rainfed mode and other practices (Bégué et al., 2018). It is thus necessary to adapt the classification approaches (satellite data and algorithms as well as training and validation in situ data) to the large variability of the farming systems in the world (Dixon et al., 2001), and thus to have access to appropriate training data.

The arrival of Sentinel-1 and Sentinel-2 satellite image time series, the emergence of new classification algorithms in the domain of machine learning and artificial intelligence, and easy access to preprocessed images and image processing tools on
web platforms have democratized image processing and opened up new avenues for LULC mapping over large areas. Following this trend, large benchmark datasets acquired using annotation tools of satellite images all over the world have multiplied to train algorithms and validate remote sensing-derived products (Long et al., 2020). However, these datasets have a broad LULC nomenclature, and agricultural land use is often reduced to a single class due to difficulties in discriminating

cropping practices from satellite images. The main data sources currently available for agricultural land use mapping in southern countries are listed below.

At a global and continental scale, initiatives that freely distribute land cover reference datasets exist (see review by Tsendbazar et al. (2015)). The GOFC-GOLD (Global Observation for Forest and Land Cover Dynamics; see http://www.gofcgold.wur.nl/sites/gofcgold_refdataportal.php for further details and access to data) regroups and consolidates existing reference datasets used for the validation of legacy global land cover products (prior to 2015) at moderate spatial resolution (300m-1km) such as GLC 2000 and GlobCover 2005. All referenced databases are provided at global scale, ranging from few hundreds to around 2,000 samples each. Except for GlobCover 2005, which contains a "rainfed cropland" class, other referred LC nomenclatures only contain a single cropland class, sometimes referred to as "cultivated".

Other data collection experiences reached a sensibly higher number of samples through the use of crowdsourcing campaigns, a notable example being the LULC reference dataset presented in Fritz et al., 2017, and its companion work from Laso Bayas et al., 2017b: thanks to the Geo-Wiki tool providing an easy-to-use interface for the photointerpretation of very high spatial resolution satellite images, it was possible to collect up to 150 000 samples of different LULC classes. This includes over 36000 cropland locations, distributed over contrasted areas in terms of cropland density. As in the previous case, a single cropland class is referenced in the nomenclature, alone or mixed with natural vegetation ("mosaic" class). Although crowdsourcing confirms as a valuable strategy to collect reference cropland data at larger scales, it still remains unsuited when precise information has to be collected, both spatially (resolution, plot boundaries, etc.) and in terms of crop type nomenclatures. Matter of facts, most of the crowdsourcing initiatives are based on visual image interpretation, which prevents the precise localization and identification of cropping practices. Shifting to a crowdsourced field strategy will not be suitable as well, both because of the very specific agronomic and GIS competences needed and the limited accessibility to cultivated areas in tropical countries.

Recently, the LandCoverNet dataset was released for the African continent (Alemohammad et al., 2020), with the specific aim to foster the use of recent machine and deep learning approaches for automatic land cover classification. Here, samples are provided in the form of densely annotated image chips (256x256 pixels at 20m resolution) accompanied by the corresponding Sentinel-2 observations over the reference year (2018). A total number of 1 980 fully annotated chips, accounting for more than 30 million labeled pixels, are provided, spanning 66 tiles of Sentinel-2 over the entire African continent. Although such dataset could allow a finer spatial validation of LULC products at high resolution, it still provides a single "cultivated land" class, making it unsuitable for the assessment of LULC products specifically conceived for the monitoring of agricultural systems.

These data are used to validate global (Hoskins et al., 2016) or national cropland maps (Laso Bayas et al., 2017a) as the nomenclature used for labeling the classes does not specify the crop type.

At a national scale, ground campaigns, such as those carried out as part of the Sen2Agri project in South Africa and Mali, collected data on the main crop types (Defourny et al., 2019). However these data are generally not available to validate global maps or train new classification algorithms, as they are often the responsibility of national sovereignty.

At a local scale, datasets on crop types have been acquired, and are still acquired, across multiple world regions within the context of the JECAM (Joint Experiment for Crop Assessment and Monitoring; Available online: http://www.jecam.org/; 105    accessed on 10 February 2020) international network. The JECAM initiative was first developed under the GEO (Group of Earth Observations) umbrella and then became the research and development component of GEOGLAM (GEO Global Agricultural Monitoring), to enable the global agricultural monitoring community to carry out cross site experiments and compare results based on disparate sources of data, using various methods, over a variety of local or regional cropping systems. Data are acquired following a given protocol and nomenclature (see Defourny et al. (2014)). The experiment has been operating 110    since 2013, and some in situ datasets produced at the field scale have been used in different benchmarking mapping studies (Waldner et al., 2016; Inglada et al., 2015). However, only a part of the collected ground data was used in these studies and the databases are not publicly shared.

To make agricultural land use data publicly available to the remote sensing community, for classification algorithm benchmarking or LULC product validation for example, an important work of harmonization of in situ JECAM and JECAM- 115    like agricultural land use datasets was undertaken for nine sites located in the tropical belt. The acquisition protocol was adapted from Defourny et al. (2014) to take into account the characteristics of tropical agriculture (e.g., small field size, accessibility). At each site, information on crop type and cropping practices was collected locally, at the field level, with a detailed nomenclature. The acquisition period was between 2013 and 2020, and the number of monitoring years per site was between 1 and 7.

In this paper, we describe in detail the study sites, the data collection protocol and the structure of the final database. We then discuss how the harmonization of the dataset and the diversity of the studied agrosystems, including small-holder farming, make our dataset unique and valuable for applications in emerging/developing countries in the tropics.

 **2. Methods**

**2.1 Study sites**

Except for Cambodia, the study sites belong to the JECAM network (http://www.jecam.org/), and cover several hundred square kilometers each. The nine sites are spread over seven countries of the tropical belt (Figure 1) and cover different farming systems (Figure 2).

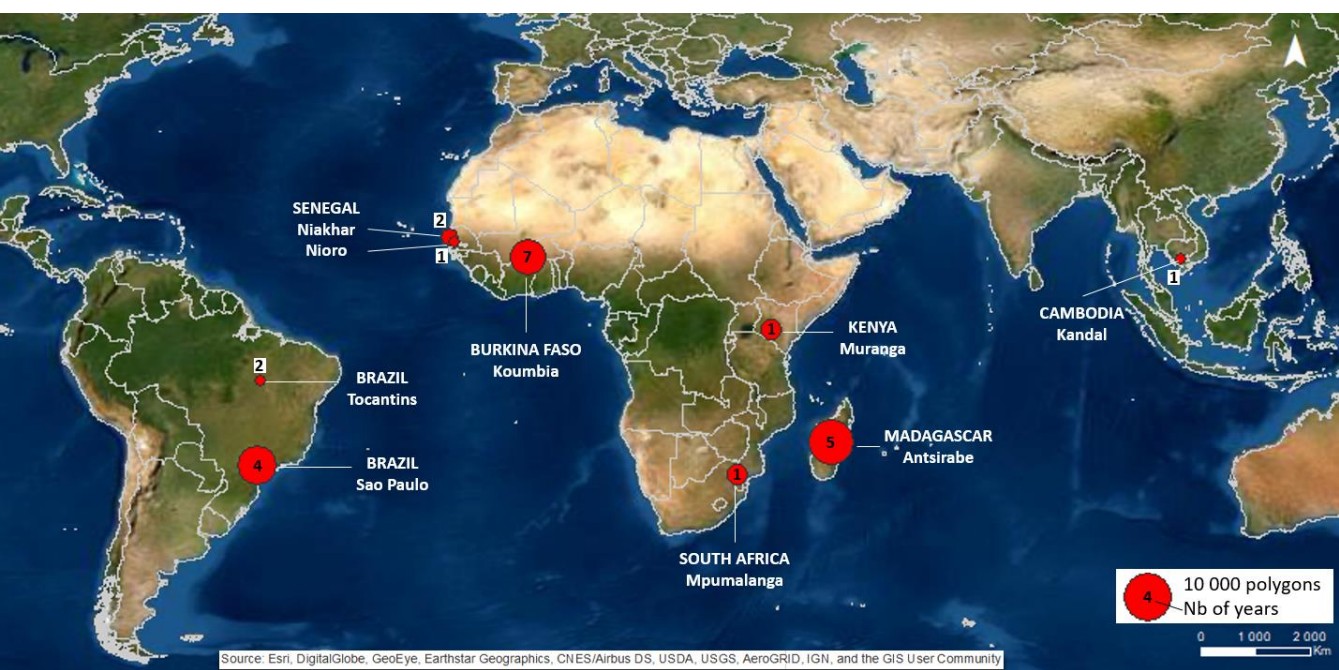

**Figure 1. Location map of the study sites, and the associated number of collection years and sampled plots (symbolized by the size of the red circles)**.

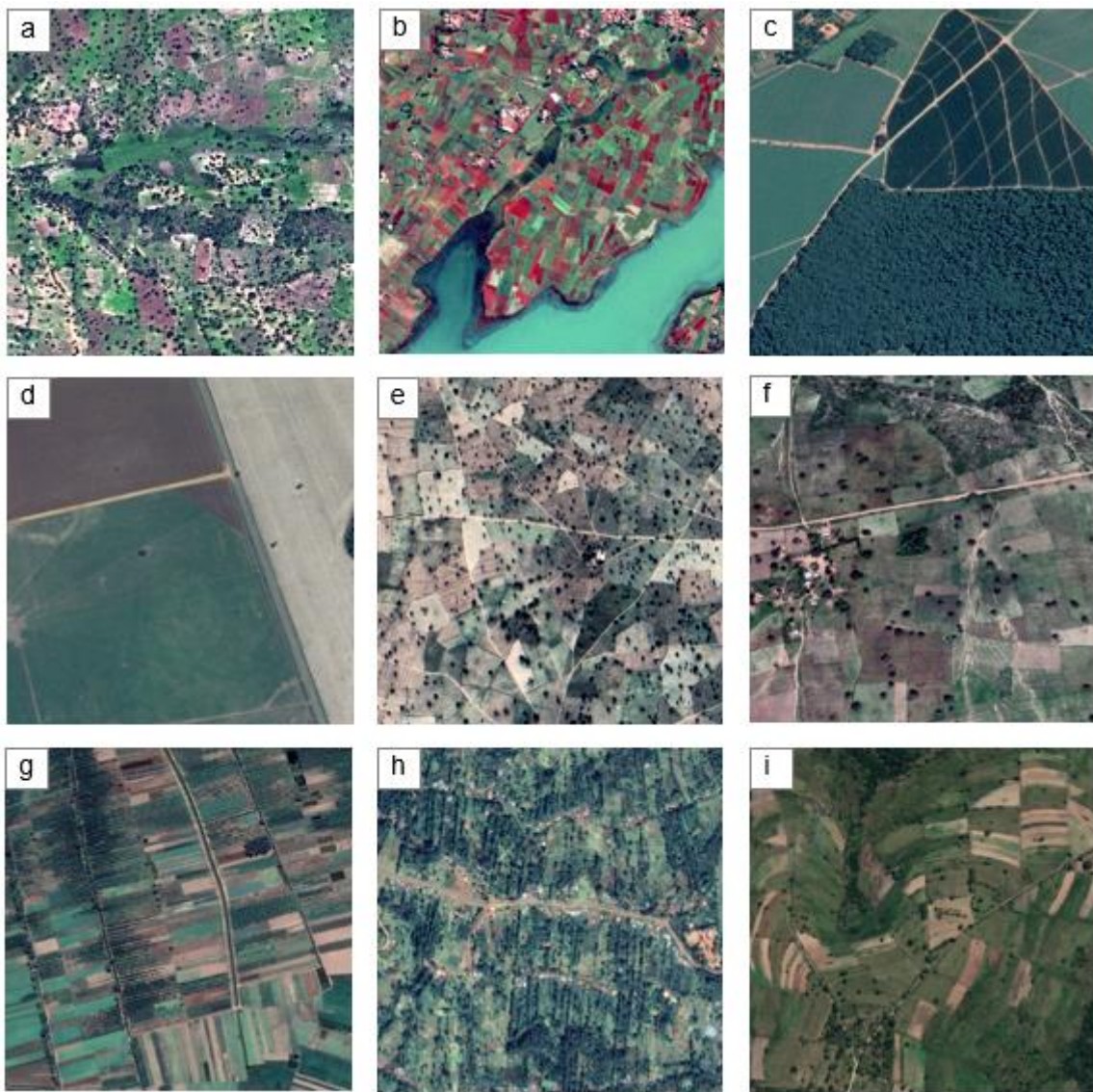

**Figure 2. A 1 km² sample of land showing the landscape variety across the sampled sites due to the farming system in place** : (a) rainfed cereals in Burkina-Faso; (b) rice systems in Madagascar; (c) agropastoral systems in Brazil-Tocantins; (d) mixed agriculture in Brazil-São Paulo; (e) rainfed groundnut and millet agropastoral systems in Niakhar and (f) in Nioro, Senegal; (g) irrigated rice systems and orchards in Cambodia; (h) agroforestry in Kenya; (i) mixed agriculture in South Africa. *Images* © **Google Earth 2020**

The JECAM Burkina Faso study site is a 60 x 60 km area located around the town of Koumbia, Tuy Province, in the southwest of the country. The climate is tropical. The absence of significant relief and the relatively good conditions in terms of soil and climate favored the densification of cropped surfaces, which span the majority of the area: arable lands cover more than 60% of the site, and the remaining surface is either unsuitable for cultivation (e.g., rocky) or protected areas for nature conservation. The landscape is characterized by an alternation of large cropland areas made up of a patchwork of diversified small cropped

fields (approximately 1 ha) and areas covered by natural vegetation. With the exception of few lowland rice plots, all crops are rainfed and hence cultivated during the rainy season that occurs from May to October (approximately 1000 mm average annual rainfall). The main crops are more or less equally distributed between cash crops (mainly cotton) and staple crops, with a significant predominance of cereal crops (maize, sorghum, millet, and locally rice) over oleaginous (sesame, groundnuts) and leguminous (peas/cow peas, soybeans) crops.

The JECAM Madagascar study site is a 60 x 60 km zone located in the Vakinankaratra region, around Anstirabe city, in the central highlands of the country. It is characterized by terraced mountainous terrain at 1200 to 1500 m of altitude, rice-growing valleys positioned between grassy hills and rocky outcrops. The climate is subtropical, with a rainy season from December to February. The average annual precipitation is 1300 mm. The growing season occurs from October to June. Cultivated crops are diversified, although maize and rice predominate. Fruit production is also present in the area. The mean size of an agricultural field in the area is very small (approximately 0.05 ha), but contiguous fields of the same crop type occasionally give rise to larger single crop patches. Rice is mainly grown in irrigated areas but has recently mingled with other rainfed crops on slopes (called tanetys). Other main crops are carrots, potatoes, sweet potatoes, soybeans or cassava.

The JECAM São Paulo site in Brazil is a large area of 90 x 130 km located in the São Paulo State, close to Botucatu city. It is composed of a relatively smooth relief with slopes mostly <5%. The region is classified as subtropical humid-dry in the winter. The average temperature is 19°C and the average annual precipitation is 1400 mm with a rainy season from December to March. The area is diversified and can be divided into four main agricultural subregions: (1) in the South-West, annual crops (maize, wheat, soybean) including summer (growth cycle from October to May) and winter crops (June to September) - some of them irrigated with center pivot systems; (2) in the Center forest plantations for wood production; (3) in the East pastures, and (4) in the North sugarcane, which has variable planting and harvesting dates: the first sugarcane cycle occurs between September and March, and is grown for approximately 12–18 months. Sugarcane reaches maximal growth in April, in this region. After the first harvest, the cycle of the ratoon sugarcane starts, with the annual cut between April and December. Natural forests, mostly along rivers, and orange orchards are present in these four subregions. The field size is generally larger than 10 ha, and can reach more than 200 ha for pastures and forest plantations. A detailed description of this site, including crop and rotation descriptions, is given in de Oliveira Santos et al. (2019).

The JECAM Tocantins site in Brazil is part of the MATOPIBA (Maranhão, Tocantins, Piauí and Bahia) region, a new agricultural frontier in Brazil. It is a 25 x 25 km site situated in the Municipality of Pedro Afonso and surroundings, in the Cerrado biome. The climate is tropical, with a rainy season from October to March. The landscape is composed of a mosaic of large fields (generally approximately 100 ha), native forest remnants and rangelands, with mild relief, and the annual rainfall is between 1700-1800 mm. The main agricultural systems are soybean single cropping, double cropping of summer soybean from November to February followed by a cereal crop (maize, millet or sorghum) from March to June, some sugarcane, and

175 planted pastures that are increasingly being implemented in the region as part of integrated crop-livestock systems (soybean-corn-planted pasture). Sugarcane crops are irrigated with center pivot systems.

The Niakhar and Nioro Senegalese study sites are located in the Senegalese Peanut Basin, in the central western part of the country. The Niakhar site spans the districts of Fatick and Bambey in the northern part of the Peanut Basin, and the Nioro site is located in the district of Nioro du Rip at the border of the Gambia. Each site covers approximately 400 km². The climate is

180 Sahelo-Sudanian with one rainy season (400 to 600 mm) that lasts from July to October. The relief is relatively flat. As in many parts of the Sahelian zone, smallholder farming systems are dominated by tree-based agricultural landscapes, forming so-called parklands. The Niakhar site is dominated by Faidherbia albida trees, while the Nioro site is dominated by Cordyla pinnata trees. The livelihoods of rural populations are centered on small-scale rainfed agriculture, with low usage of mineral fertilizer. Pearl millet and groundnut are the main staple crops mainly cultivated in biennial rotation. Other crops are sorghum,

cowpea, bissap and maize cultivated during the rainy season.

The JECAM Kenya study site is a 25 x 10 km area located approximately 50 km north of Naïrobi, including Kangema and Muranga towns, in the central province of Kenya. It is settled in a very hilly landscape with steep slopes and strong local relief variations in a general toposequence trend following an east-west altitude gradient from 1000 m to 2800 m. The climate is wet tropical, somewhat temperate by altitude and regularized by two rainy seasons (from March to May or June and from October

to November) with 1200 to 2000 mm annual rainfall depending on the altitude. The permanent moisture and good natural drainage of a rich volcanic loam allows for intensive agriculture, mainly based on perennial crops (mostly banana, various fruits, coffee, and tea) associated with dairy farming and rainfed horticultural as well as food crops (e.g., French beans, cabbage, maize, cassava). The latter are cropped all year long, except in January and July which are dry months, and without a defined seasonal calendar (maize, for instance, can have three cycles per year). The mean size of an agricultural field in the

area is very small (approximately 0.08 ha), resulting in a patchwork landscape of heterogeneous fields with a great diversity of structures.

The Cambodia study site corresponds to a 30 km radius buffer area around Wat Pi Chey Saa Kor, Kom Poung Kor village, Kandal Province, where the ecology of fruit bats Pteropus lylei was recently investigated (Choden et al., 2019). The area is characterized by a tropical climate with a rainy season from May to October. The annual rainfall is between 1000 and 1500

200 mm. Two main rivers, the Mekong and the Bassac, cross the area. In this flat region, rice is the dominant crop, mainly grown in irrigated areas from May to October. Fruit plantations (mango, sapodilla) and natural wetlands are also present. The mean field size is small (approximately 1 ha). The population lives in villages along roads composed of small houses with fruit tree backyards.

The JECAM South African study area is a 60 x 60 km site located in Mpumalanga Province in the northeastern part of the

205 country, close to the Mozambique border corresponding mostly to a subsistence agriculture area. The climate is subtropical

with a rainy season from November to February. The annual rainfall is between 600 and 800 mm. The site is characterized by a bush-clad plain between the Drakensberg Mountains (West) and savannahs (East) with several wildlife reserves (e.g. Kruger Park). The study area is characterized by smallholder agriculture (generally less than 1 ha), with diversified crops: cereals, groundnuts, potatoes, vegetables and fruit crops. Important timber plantations are present on the western part of the site.

**2.2 Data collection**

The acquisition protocol is based on the JECAM guidelines (Defourny et al., 2014) with adaptations to consider some characteristics of tropical agriculture (mainly small field size and accessibility). Field surveys were conducted at least once in each study zone, with several sites revisited over multiple consecutive years (up to 7 for the Burkina Faso site). Campaigns took place either around the growing peak of the cropping season, for the sites with a main growing season linked to the rainy season such as Burkina Faso, or seasonally, for the sites with multiple cropping (e.g. São Paulo site). Except for Senegal where a stratified sampling plan for field surveys was used (Ndao et al., 2021), the GPS waypoints were gathered following an opportunistic sampling approach (called the "windshield survey") along the roads or tracks according to their accessibility (which can be difficult during the rainy season, leading to fewer surveys on secondary roads or tracks in some study areas) while ensuring the best representativity of the existing cropping systems in place (Defourny et al., 2014; Waldner et al., 2019). GPS waypoints were also recorded on different types of noncrop classes (e.g., natural vegetation, settlement areas, water bodies) to allow differentiating crop and noncrop classes. Waypoints were only recorded for homogenous fields/entities of at least 20 x 20 m² (against a minimum sampling unit of 0.25 ha with a minimum width of 30 m in JECAM guidelines). To facilitate the location of sampling areas and the remote acquisition of waypoints, field operators were equipped with GPS tablets (Trimble - Yuma2 or Handheld - Algiz 10X) providing access to a QGIS project with Very High Spatial Resolution (VHSR) images (orthorectified Pleiades or SPOT 6/7 images ordered just before the surveys, or PlanetScope images). This equipment allowed the in situ recording of attributes relative to each waypoint on data entry forms (with the automatic filling of IDs or dates and scrollable lists for other attributes to avoid data entry errors (Figure 3.a and Appendix A))). For each waypoint, a set of attributes, corresponding to the cropping practices (crop type, cropping pattern, management techniques) were recorded. An attribute referred to as "Keywords" was also created to associate various generic terms (land cover, crop group, crop type, cropping practice, etc. (Appendix B)) to each polygon. This attribute has two objectives: (i) facilitating keyword search for the user and (ii) allowing the user to create his own nomenclature (hierarchical or not) with different levels of detail so that the nomenclature can be dedicated to the user's needs. These terms are based on the FAO land use definitions (FAO, 2020) and JECAM hierarchical nomenclature (Defourny et al., 2014), which were adapted to take into account the diversity of the farming systems in the surveyed sites. All these attributes are described in Table 1.

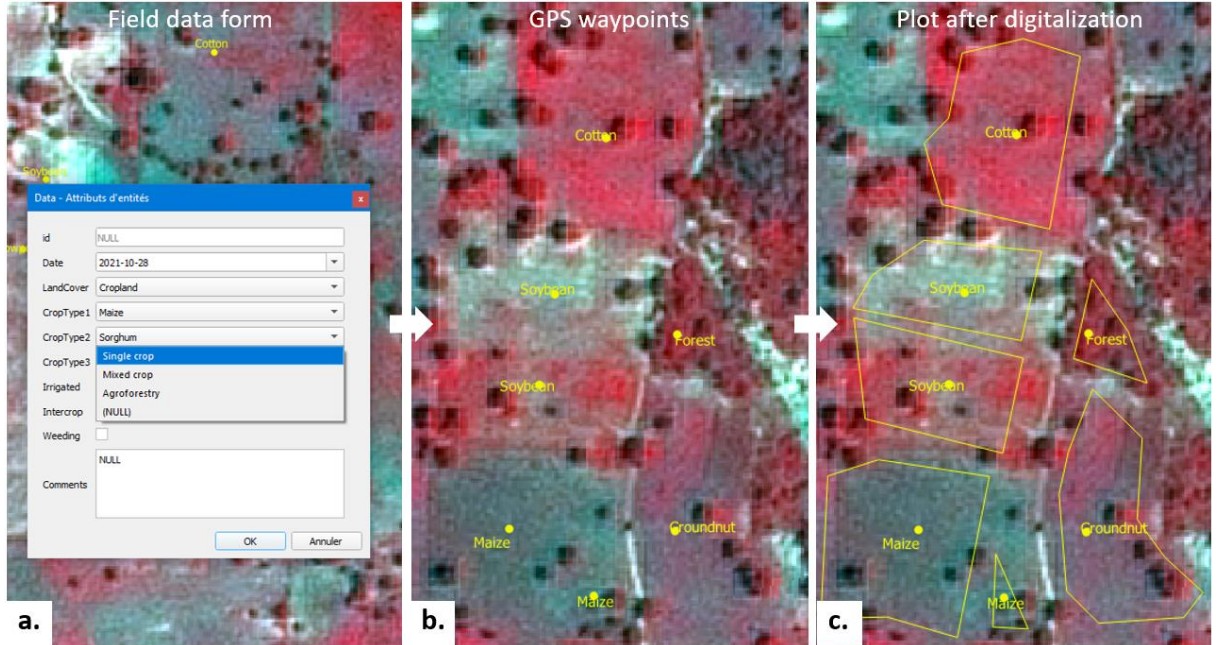

**Figure 3. Workflow of the data acquisition:** (a) field data form used on the GPS tablet, (b) GPS waypoints acquired in the field and (c) corresponding plots after digitalization of the boundaries, displayed on a satellite image in false color (Red: Near Infrared band, Green: Red band, Blue: Green band).

In the specific case of the Burkina Faso, Senegal-Niakhar and Brazil-São Paulo sites, the same fields were revisited each year to study crop rotations and fallow practices in the region. For the South African site, some points were collected by helicopter using the Producer Independent Crop Estimates System (PICES (Fourie, 2009)) method developed by the National Crop
Statistics Consortium. Flights were performed at an average altitude of 500 feet and a low flying speed, allowing us to record GPS points and to determine land use using a GPS tablet associated with a GIS interface and a recent VHRS image. Only clearly identifiable land covers were kept in the database.

During a field mission, the team is composed of an agronomist with geoprocessing skills, accompanied by a national researcher or technician with expertise in the local farming systems and a local driver. In some countries (Burkina Faso, Senegal,
Madagascar, Kenya), local partners were trained to collect data. The training sessions were carried out directly in situ to be as close as possible to reality. The data acquisition duration varies in many of the visited areas: in Brazil (large fields and good road infrastructures), 300 plots can be visited in one day while for other sites (small to very small fields), it is possible to collect between 50 and 150 plots per day (depending on the road state and field accessibility). Usually, the mission for a 3600 km² site of smallholders is one week with approximately 700 plots visited.

**2.3 Postprocessing**

Once the waypoints were acquired (Figure 3.b), the boundaries of each field or noncrop entity were digitized on the VHSR images in the QGIS software, and the class labels (and other attributes, see Table 1) were attached to the polygon database (Figure 3.b). Additional noncrop polygons were added by CAPI (computer-assisted photo interpretation) of the VHSR images for the built-up areas, water bodies, wetlands, mineral surfaces, and natural forest classes (land covers clearly identifiable on images).

To avoid digitizing errors, this step was performed by the same operator as the one who performed the field surveys. Despite this, if there was doubt on the delineation of a given entity (e.g. fuzzy boundaries, high heterogeneity), the given entity was removed from the database. Finally, the topology of each entity was controlled externally.

**3. Data Records**

This database, which contains 27 197 records, is a geographic layer in Shapefile format. Each record corresponds to a polygon with 16 attributes (Table 1). Because of the dispersion of study sites on the globe, the layer is in a geographic coordinates system with Datum WGS84. The distribution of the different records over the study sites is reported in Table 2, along with information on the temporal (corresponding years) and spatial coverage (source, number and average size of digitized polygons).

Twenty different land cover types and 102 different crop types were observed. More than ¾ of the observations are agricultural land, and the most represented crop types are maize, rice and sugarcane. The distributions of the main land cover and crop types are represented in Figures 4 and 5. Figure 6 summarizes the distribution of the data acquisition method by site and shows that 87% of the data come from an in situ survey.

| Attribute Name | Data Type | Description / available arguments | Example |
|---|---|---|---|
| Id | Numeric | Unique ID | *26413* |
| Country | Text | Country name | *Burkina Faso* |
| SiteName | Text | Site name (generally related to the biggest city around or to the region name) | *Koumbia* |
| DataSource | Numeric | Discrimination between land uses acquired from *in situ* surveys or satellite image CAPI (computer assisted photointerpretation)<br>0: Land use from *in situ* survey<br>1: Land use from satellite image interpretation<br>2: Land use from aircraft observation | *0* |

| AcquiDate* | Date | *In situ* survey acquisition date (when the land use is photointerpreted, see "DataSource" attribute) – Format: yyyy-mm-dd | *2020-10-21* |
|---|---|---|---|
| LandCover | Text | Land cover of the polygon. If value is "Cropland", see CropType 1, 2 and 3 attributes for more information | *Cropland* |
| CropType1 | Text | Main crop type of the polygon | *Cotton* |
| CropType2 | Text | Secondary crop type of the polygon (in case of intercropping) | *Maize* |
| CropType3 | Text | Tertiary crop type of the polygon (in case of intercropping) | *NULL* |
| SOS* | Date | Start of season date in the site (if empty, this means that no specific season exists in the study area) – Format: yyyy-mm-dd | *2020-05-01* |
| EOS* | Date | End of season date in the site (if empty, this means that no specific season exists in the study area) – Format: yyyy-mm-dd | *2020-11-30* |
| Irrigated | Numeric | Presence/absence of an irrigation system<br>  0: No information available<br>  1: Rainfed<br>  2: Irrigated<br>  Empty: For polygons other than cropland | *1* |
| Intercrop | Numeric | Presence/absence of intercropping<br>  0: Single crop<br>  1: Mixed crop or row intercrop<br>  2: Agroforestry<br>  Empty: For polygons other than cropland | *1* |
| Weeding | Numeric | Presence/absence of weeds<br>  0: No information available<br>  1: Presence of weeds<br>  Empty: For polygons other than cropland | *0* |
| Area_ha | Numeric | Polygon area in hectares | *0.446* |
| KeyWords | Text | Set of terms associated with the land use of the polygon (separated by semicolons ";") | *Agricultural land ; Cropland ; Arable land ; Temporary crop ; Cash crop ; Fiber crop* |

\* For each field at the Tocantins site, the operator was able to record the crop type for the two cropping seasons by observing the crop residues in the field or by interviewing the farmers. Consequently, the acquisition date of those polygons does not

always correspond to the actual land cover of the field. The user must refer to the SOS and EOS dates to identify the season corresponding to the crop type recorded.

**Table 1. Description of the attributes recorded for each polygon of the database.**

| Country, Site name | Number of collection years | Total number of polygons (percentage of crop polygons in the dataset) | Mean polygon size (ha)* | Percentage of polygons obtained from ground survey | Nb. of crop type classes |
|---|---|---|---|---|---|
| **Burkina Faso, Koumbia** | 7 (2013 to 2020) | 6 264 (79%) | 0.60 | 89% | 23 |
| **Madagascar, Antsirabe** | 5 (2015 to 2019) | 8 351 (87%) | 0.35 | 95% | 47 |
| **Brazil, São Paulo** | 4 (2014 to 2017) ** | 6 149 (66%) | 22 | 96% | 21 |
| **Brazil, Tocantins** | 2 (2015 and 2016) | 533 (56%) | 150 | 67% | 7 |
| **Senegal, Niakhar** | 2 (2018 and 2019) | 1 403 (74%) | 0.54 | 83% | 5 |
| **Senegal, Nioro** | 1 (2018) | 457 (46%) | 1.17 | 48% | 6 |
| **Kenya, Muranga** | 1 (2015) | 1 647 (77%) | 0.14 | 100% | 26 |
| **Cambodge, Kandal** | 1 (2014 / 2015) | 529 (25%) | *** Small fields | 28% | 5 |
| **South Africa, Mpumalanga** | 1 (2017) | 1 741 (59%) | *** Small fields | 38% | 10 |

\*      Areas calculated on cropland polygons
\*\*     16 field campaigns in 4 years
\*\*\*      The digitized boundaries of the polygons correspond to homogeneous crop areas (collections of adjacent small fields) and not necessarily to single fields.

**Table 2. Synthetic view of the final GIS database.**

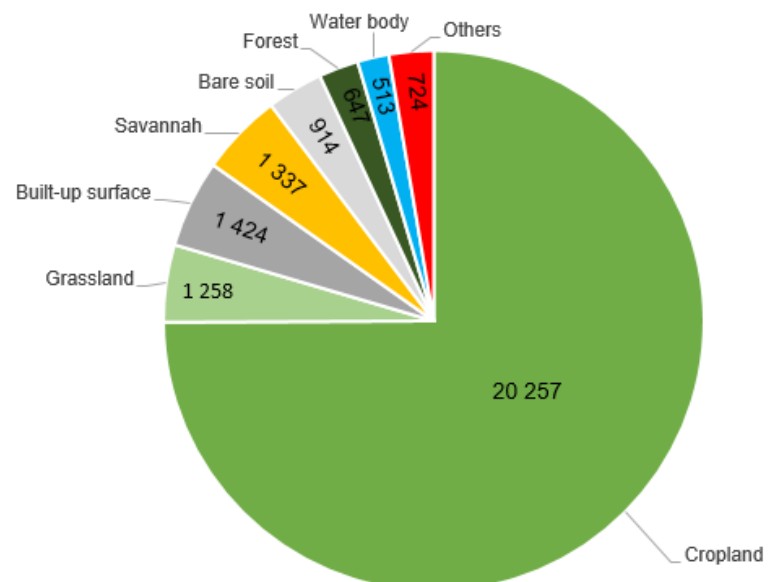

**Figure 4. Distribution of the main land cover types (in number of polygons).**

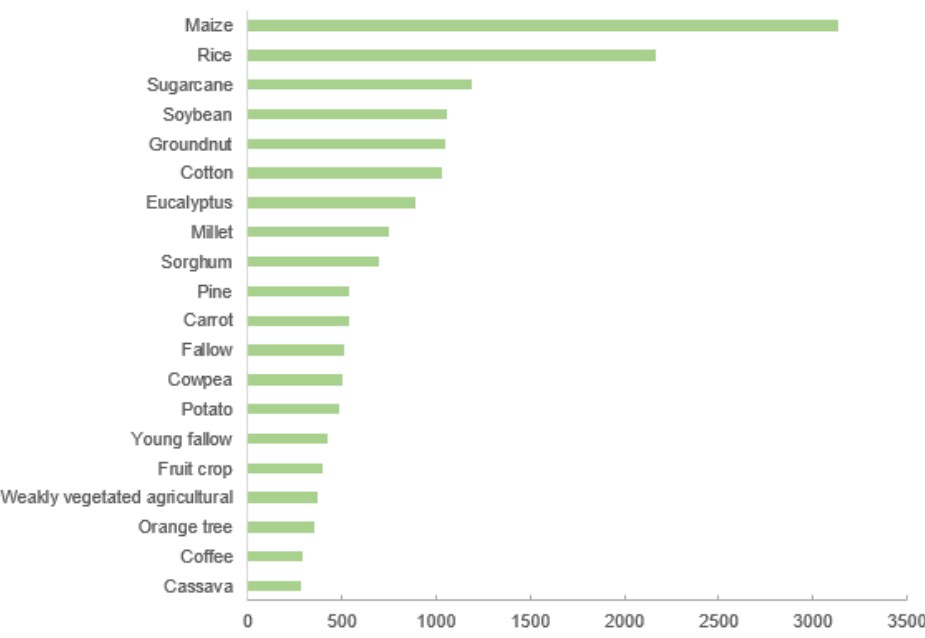

**Figure 5. Distribution of the main crop types (in number of polygons).**

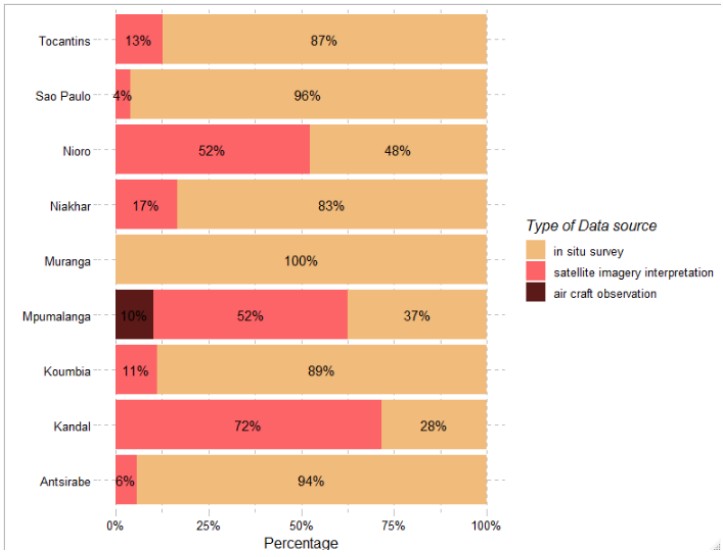

**Figure 6. Distribution of the data sources, given in percentage of the total number of polygons per site.**

## 4. Technical Validation

### 4.1 Quality Checking

Due to the nature of the dataset (in situ observation), validation is not possible. However, quality control was performed throughout the data chain, from acquisition to postprocessing, to ensure the quality of the datasets and their homogeneity throughout the sampled years and locations.

First, the acquisition protocol was described in a technical guide provided to the field teams so that nothing was forgotten during the campaigns. The dropdown list in the data entry form reduced input and postprocessing errors.

Second, during the postprocessing step, the orthorectification of the VHSR images used to digitize the fields was checked from one year to the next, for multiyear sites and corrected if necessary by taking homologous points. The fields were then manually digitized on the VHSR images, and the photographs taken in situ were used whenever necessary. In the case of doubtful data, these data were discarded and removed from the dataset.

Finally, each site has a referee person who knows the area very well. He supervises the entire chain from data collection to database integration. In this way, each step is conducted by a specialist (agronomy, GIS, database) in complementarity with the referee to minimize errors and contribute to the overall quality of the datasets.

## 4.2 Representativeness of datasets

Because of their small size, these sites cannot be considered representative of the entire country in which they are located; however, they are claimed to be representative of an area that encompasses more than the JECAM site. To specify the extent of this representative area, we referred to existing zoning maps. We used the two reference maps available for Southern countries: the FEWS-NET livelihood zones map (https://fews.net/fews-data/335) and the FAO farming systems map (http://www.fao.org/farmingsystems/mapstheme_01_en.htm). The livelihood zones are produced at the national scale and are available for 38 developing countries. The zones are defined as geographical areas within which people broadly share the same patterns of livelihood (i.e., broadly the same production system, the same income earning opportunities and patterns of trade) (see Grillo and Holt (2009) for more details). Farming system maps are available for the Global South (covering 130 countries). The classes are defined as a population of individual farm systems that have broadly similar resource bases, enterprise patterns, household livelihoods and constraints (Dixon et al., 2001; Auricht et al., 2014).

Although these two maps were not produced for the same purposes, they were derived using similar criteria (agro-climatology, elevation, landscape, dominant pattern of farm activities, etc.) that are closely related to agricultural land use, as recorded in the database. For both maps, the type and extent of the zones corresponding to our JECAM study sites are given in Table 3.. Unfortunately, livelihood maps are available only for four of the JECAM countries presented here.

| Country | Livelihood zone (FEWS-NET) | | Farming systems (FAO) | |
|---|---|---|---|---|
| | Livelihood type (year of production)) | km² | Farming system type (year of production) | km² |
| KENYA | Central Highlands, High Potential Zone (2011) | 19 689 | Maize mixed (2014) | 615 593 |
| MADAGASCAR | Ankaratra: staple crops, horticulture, milk (2017) | 15 675 | Rice-tree crop (Maize-mixed) 2014 | 308 489 |
| SENEGAL | Rainfed groundnut and millet (2015) | 10 256 | Agro-pastoral millet/sorghum (2014) | 1 238 113 |
| SENEGAL | Rainfed groundnut and cereals (2015) | 22 087 | Agro-pastoral millet/sorghum (2014) | 1 238 113 |
| BURKINA | West cotton and cereals (2014) | 35 813 | Cereal-root crop mixed (2014) | 1 931 654 |

| SOUTH AFRICA | | | Large commercial and smallholder (Maize-mixed or Perenial mixed) (2014) | 1 010 746 |
|---|---|---|---|---|
| BRAZIL (SP) | | | Intensive mixed (2001) | 812 259 |
| BRAZIL (TO) | | | Extensive mixed (Cerrados & Llanos) (2001) | 1 744 804 |
| CAMBODGE | | | Low-land rice (2001) | 526 678 |

**Table 3**: **Agricultural types and extent of study sites' belonging zones: FEWS-NET livelihood zones** *(source:* https://fews.net/fews-data/335*)* **and FAO farming system zones** *(*http://www.fao.org/farmingsystems/mapstheme_01_en.htm*)*.

With a mean size of the zone being approximately 20 000 km² (Table 3), we are confident that our JECAM sites are
representative of the livelihood zone to which they belong. The datasets presented here can thus be used to train or validate
land cover maps of the corresponding zones. The farming system zones are much larger (between 300 000 km² and 2 Mkm²)
and include a larger diversity of environmental and farming conditions; in these conditions it is not possible to argue that the
JECAM sites are representative of such large areas; thus, the JECAM datasets need to be completed with other datasets
belonging to the same farming system class before being used for training land cover classification algorithms. However, they
can still be used for algorithm/product validation or comparison.

It is also important to mention that other agroecological zoning (AEZ) can be used (even if only in a few areas directly related
to the agricultural land use) or that each user can produce their own AEZ and use it to delineate the area in which the JECAM
dataset can be used to train classification algorithms.

**5. Dataset application study cases**

The in situ JECAM dataset and its derived land use/land cover products have been used in a wide spectrum of studies covering
several aspects linked to agricultural monitoring attesting to the good quality of the dataset and good spatial representativeness
of tropical country farming systems.

First, specific site studies have been conducted to test several methodological aspects. For instance, land use maps combining
a supervised object-based approach with multisource high spatial resolution time series were developed in Madagascar
(Lebourgeois et al., 2017) and in Brazil (de Oliveira Santos et al., 2019). The brazilian site (São Paulo) was also included in a
broader study presenting an intercomparison of several cropland mapping methodologies over 5 contrasting JECAM sites
(Brasil, Ukraine, Russia, Argentina and China) in terms of growing conditions, characteristics and cropping practices (Waldner
et al., 2016). Very recently, following the rapid dissemination of up-to-date artificial intelligence approaches, Gbodjo et al.

(2020) and Ienco et al. (2020) proposed testing the potential of deep learning architectures for land cover mapping in Senegal
(Niakhar) and Burkina Faso, respectively.

Second, in situ data coming from the Burkina Faso site and the Madagascar site were included as test sites in the Sen2-Agri system. The Sen2-Agri system is an operational processing system that provides several agricultural products from Sentinel-2 and Landsat-8 time series during the cropping season. The two sites have been included in preliminary studies preparing the Sen2-Agri system processing chain (Bontemps et al., 2015; Valero et al., 2016), while the Madagascar site was considered
later in the demonstration phase of the system at the local scale (http://www.esa-sen2agri.org/system-demonstration/).

Last, the different in situ data and the derived products have been used in studies covering different aspects of agricultural monitoring. For instance a semiautomated clustering approach has been proposed for cropping system mapping over the Tocantins region in Brazil (Bellón et al., 2018). Using the land use map derived from the Burkina Faso site and the Senegal site (Niakhar), remote sensing-based statistical crop yield models have been proposed for maize (Leroux et al., 2019) and pearl
millet (Leroux et al., 2020b). Based on the land use map derived from the Niakhar and Nioro sites in Senegal, Ndao et al. (2021) proposed an approach to characterize the agricultural landscape heterogeneity in agroforestry parklands, which was then used to analyze how far agricultural landscape diversity contributes to the household food security (Leroux et al., 2020a).

## 6. Data availability

The dataset is ready for use on any GIS software, and can be filtered by region, year or key words. It is distributed with a CC-
370 BY license. The database, as well as the Kmz file locating the study areas, are available online on the CIRAD DataVerse at
https://doi.org/10.18167/DVN1/P7OLAP (Jolivot et al., 2021).

## 7. Conclusion and perspectives

The accurate mapping of cropland and associated cropping practices in smallholder farming systems of tropical countries is crucial for the improvement of agricultural monitoring systems at local and/or global scales. The essential prerequisite to reach
such objectives is to have available in situ datasets representative of the diverse agricultural practices in tropical countries. This paper presented a harmonized in situ crop type dataset acquired between 2013 and 2020 over nine sites spread over seven tropical countries. This dataset collected in the framework of the JECAM initiative is unique and very valuable because it is produced at the field scale, based on in situ observations and quality-controlled, and standardized observations for various tropical cropping systems, including small-holder farming systems. These characteristics allow this dataset to be used as a
benchmark to assess the performances and robustness of newly developed classification algorithms for cropland and crop type/practice mapping in diverse and documented agricultural conditions. In addition, this dataset can also be used to validate the cropland class of existing global or national LULC products, in particular those recently produced with Sentinel/Landsat image time series, and some crop type and practice (fallow, double cropping) classes. In the end it should be part of publicly

online datasets and algorithm sharing platforms as promoted by the JECAM network and Long et al. (2020) who encourage the sharing of datasets for remote sensing applications, and more broadly to the scientific community, land use planners and agricultural monitoring agencies.

Thanks to ongoing projects and funded initiatives in which our team is involved, we will provide updates to the presented dataset on a regular basis. To date, several field campaigns are already planned on some of the presented sites, and projects are being built which will lead to the inclusion of multiple new ones. Moreover, since the paper also proposes a set of technical guidelines to integrate the database, opening to external contributors may lead to a significant extension of the geographic coverage of the database, and hence of its representativity with respect to the diversity of tropical agrosystems. As future work, we intend to carry out a study about the development of a technical solution aimed at facilitating such external contributions (e.g., a compliant data collection tool and workflow).

**Author contributions**

AJ, AB, LR and VL wrote the manuscript with substantial contributions of the PI's site: VL (Madagascar), RG (Burkina Faso), LL (Senegal), GL (Brazil – Sao Paulo), BB (Brazil – Tocantins), AT (Cambodia), CL (Kenya) and TN and AJ (South Africa)

AJ, VL and RG designed the database.

AJ harmonized and compiled the data.

Ground data collection and pre-processing : BN and MD (Senegal Niakhar), IT (Senegal Nioro), AC, ER, MA, SaD, StD, VA and VL (Madagascar), AB, AJ, CJ, DL, LL, MC, RG and StD (Burkina), GL (Brazil – Sao Paulo), AB and BB (Brazil – Tocantins), CL and MM (Kenya), AJ and MG (Cambodia), AJ and AT (South Africa)

**Competing interests**

The authors declare that they have no conflict of interest.

**Acknowledgements**

This research was mainly supported by CIRAD (scientists funding), and by various projects (CNES APR TOSCA projects, SIGMA FP7 n°603719; ESA Sen2Agri n° ESRIN 400109979/14/I-AM; CNPq n°454292/2014-7 and n°307560/2016-3; FAPESP-Microsoft Research n°2014/50715-9, the SERENA project funded by the Cirad-INRA metaprogramme GloFoodS). The SPOT and Pleiades images were acquired through the GEOSUD Program EQPX-20 funded by French National Research Agency. The PlanetScope images were acquired through the Planet's Ambassador Program.

Thanks to Embrapa Pesca e Aquicultura (Palmas, TO Brazil), Centre de Suivi Ecologique (Senegal), Agricultural Research Council (South Africa), Pasteur Institut (Cambodia), FOFIFA / DP SPAD (Madagascar), DP ASAP (Burkina), ICRAF (World

Agroforestry Center, Kenya), NIPE and FEAGRI (Brazil), and Eder Araujo da Silva (Floragro Apoio, Brazil) for their technical support.

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

## Appendix A

Example of scrollable lists used in the form. The crop type list depends on the study site (it is not necessary to mention crops not present on the site). Here is an example for the Burkina Faso site.

| Land Cover | Crop type | Irrigated | Intercrop |
|---|---|---|---|
| Bare soil | Bissap | No information | Single crop |
| Built-up surface | Cashew tree | Rainfed | Mixed crop or row inter-crop |
| Burn area | Cotton | Irrigated | Agroforestry |
| Cropland | Cowpea | | |
| Forest | Ecalyptus | | |
| Grassland | Fallow | | |
| Herbaceous savannah | Gombo | | |
| Herbaceous vegetation | Groundnt | | |
| Mineral soil | Hibiscus | | |
| Mixed trees | Maize | | |
| Pasture | Mid fallow | | |
| Savannah | Millet | | |
| Savannah with shrubs | Pea | | |
| Savannah with trees | Rice | | |
| Shrub land | Sesame | | |
| Water body | Sorghum | | |
| Wetland | Soybean | | |
| | Young fallow | | |

## Appendix B

Keywords list

| LandCover | KeyWords |
|---|---|
| Agricultural bare soil | Agricultural land ; Cropland ; Arable land ; Temporary crop |
| Albizia gummifera | Agricultural land ; Cropland ; Permanent crop ; Multifunctionnal woody crop |
| Annual crop | Agricultural land ; Cropland ; Arable land ; Temporary crop |
| Apple tree | Agricultural land ; Cropland ; Permanent crop ; Fruit crop |
| Asparagus | Agricultural land ; Cropland ; Arable land ; Temporary crop ; Vegetables and melon ; Leafy or stem vegetables |
| Asphalt road | Built-up surface |
| Avocado tree | Agricultural land ; Cropland ; Permanent crop ; Fruit crop |
| Banana | Agricultural land ; Cropland ; Permanent crop ; Fruit crop |
| Bare soil | Bare soil |
| Barley | Agricultural land ; Cropland ; Arable land ; Temporary crop ; Cereals |
| Bean | Agricultural land ; Cropland ; Arable land ; Temporary crop ; Leguminous |
| Beet | Agricultural land ; Cropland ; Arable land ; Temporary crop ; Vegetables and melons ; Root, bulb or tuberous vegetables |
| Built-up surface | Built-up surface |
| Burn area | Bare soil ; Permanent meadow and pasture ; Naturally growing |

| | |
|---|---|
| Cabbage | Agricultural land ; Cropland ; Arable land ; Temporary crop ; Vegetables and melons ; Leafy or stem vegetables |
| Cape mahogany | Agricultural land ; Cropland ; Permanent crop ; Multifunctionnal woody crop |
| Carrot | Agricultural land ; Cropland ; Arable land ; Temporary crop ; Vegetables and melons ; Root, bulb or tuberous vegetables |
| Cash woody crop | Agricultural land ; Cropland ; Permanent crop ; Cash woody crop |
| Cashew tree | Agricultural land ; Cropland ; Permanent crop ; Fruit crop |
| Cassava | Agricultural land ; Cropland ; Arable land ; Temporary crop ; Root/tuber crop with high starch or inulin content |
| Cereals | Agricultural land ; Cropland ; Arable land ; Temporary crop ; Cereals |
| Coffee | Agricultural land ; Cropland ; Permanent crop ; Cash woody crop |
| Cordia Africana | Agricultural land ; Cropland ; Permanent crop ; Multifunctionnal woody crop |
| Cotton | Agricultural land ; Cropland ; Arable land ; Temporary crop ; Cash crop ; Fiber crop |
| Cowpea | Agricultural land ; Cropland ; Arable land ; Temporary crop ; Leguminous |
| Croton | Agricultural land ; Cropland ; Permanent crop ; Multifunctionnal woody crop |
| Cucumber | Agricultural land ; Cropland ; Arable land ; Temporary crop ; Vegetables and melons ; Fruit-bearing vegetables |
| Cucurbit | Agricultural land ; Cropland ; Arable land ; Temporary crop ; Vegetables and melons ; Fruit-bearing vegetables |
| Cyprus | Agricultural land ; Cropland ; Permanent crop ; Multifunctionnal woody crop |
| Dirt track | Bare soil |
| Eucalyptus | Agricultural land ; Cropland ; Permanent crop ; Cash woody crop |
| Fallow | Agricultural land ; Cropland ; Arable land ; Fallow |
| Ficus lutea | Agricultural land ; Cropland ; Permanent crop ; Multifunctionnal woody crop |
| Forest | Natural vegetation |
| Forest plantation | Agricultural land ; Cropland ; Permanent crop ; Cash woody crop |
| Fruit crop | Agricultural land ; Cropland ; Permanent crop ; Fruit crop |
| Fruit-bearing vegetable | Agricultural land ; Cropland ; Arable land ; Temporary crop ; Vegetables and melons ; Fruit-bearing vegetables |
| Gabon tulip tree | Agricultural land ; Cropland ; Permanent crop ; Multifunctionnal woody crop |
| Goat tree | Agricultural land ; Cropland ; Permanent crop ; Multifunctionnal woody crop |
| Gombo | Agricultural land ; Cropland ; Arable land ; Temporary crop ; Vegetables and melons ; Fruit-bearing vegetables |
| Grasses and other fodder crop | Agricultural land ; Cropland ; Arable land ; Temporary crop ; Grasses and other fodder crop |
| Grassland | Agricultural land ; Permanent meadow and pasture ; Naturally growing |
| Grevillea | Agricultural land ; Cropland ; Permanent crop ; Multifunctionnal woody crop |
| Groundnut | Agricultural land ; Cropland ; Arable land ; Temporary crop ; Oilseed crop ; Leguminous ; Root, bulb or tuberous vegetables |
| Herbaceous savannah | Natural vegetation ; Grass land ; Savannah |
| Herbaceous vegetation | Natural vegetation ; Grass land |
| Hibiscus | Agricultural land ; Cropland ; Permanent crop ; Multifunctionnal woody crop |
| Jatropha | Agricultural land ; Cropland ; Permanent crop ; Cash woody crop |
| Leafy or stem vegetable | Agricultural land ; Cropland ; Arable land ; Temporary crop ; Vegetables and melons ; Leafy or stem vegetables |

| Leguminous | Agricultural land ; Cropland ; Arable land ; Temporary crop ; Oilseed crop ; Leguminous |
|---|---|
| Macadamia tree | Agricultural land ; Cropland ; Permanent crop ; Fruit crop |
| Maize | Agricultural land ; Cropland ; Arable land ; Temporary crop ; Cereals |
| Mango tree | Agricultural land ; Cropland ; Permanent crop ; Fruit crop |
| Market gardening | Agricultural land ; Cropland ; Arable land ; Temporary crop |
| Mid fallow | Agricultural land ; Cropland ; Arable land ; Fallow |
| Millet | Agricultural land ; Cropland ; Arable land ; Temporary crop ; Cereals |
| Mineral soil | Bare soil |
| Mixed annual crops | Agricultural land ; Cropland ; Arable land ; Temporary crop |
| Mixed Cereals | Agricultural land ; Cropland ; Arable land ; Temporary crop ; Cereals |
| Mixed trees | Agricultural land ; Cropland ; Permanent crop ; Fruit crop ; Natural vegetation ; Forest |
| Napier grass | Agricultural land ; Cropland ; Arable land ; Temporary crop ; Grasses and other fodder crop |
| Oat | Agricultural land ; Cropland ; Arable land ; Temporary crop ; Cereals |
| Oilseed crop | Agricultural land ; Cropland ; Arable land ; Temporary crop ; Oilseed crop |
| Old fallow | Agricultural land ; Permanent meadow and pasture ; Naturally growing |
| Onion | Agricultural land ; Cropland ; Arable land ; Temporary crop ; Vegetables and melons ; Root, bulb or tuberous vegetables |
| Orange tree | Agricultural land ; Cropland ; Permanent crop ; Fruit crop |
| Other crop | Agricultural land ; Cropland |
| Papaya tree | Agricultural land ; Cropland ; Permanent crop ; Fruit crop |
| Pasture | Agricultural land ; Permanent meadow and pasture ; Naturally growing |
| Pea | Agricultural land ; Cropland ; Arable land ; Temporary crop ; Leguminous |
| Peach tree | Agricultural land ; Cropland ; Permanent crop ; Fruit crop |
| Pear tree | Agricultural land ; Cropland ; Permanent crop ; Fruit crop |
| Pine | Agricultural land ; Cropland ; Permanent crop ; Cash woody crop |
| Pineapple | Agricultural land ; Cropland ; Arable land ; Temporary crop ; Vegetables and melons ; Fruit-bearing vegetables |
| Potato | Agricultural land ; Cropland ; Arable land ; Temporary crop ; Root/tuber crop with high starch or inulin content |
| Ravintsara | Agricultural land ; Cropland ; Permanent crop ; Multifunctionnal woody crop |
| Rice | Agricultural land ; Cropland ; Arable land ; Temporary crop ; Cereals |
| Root | Agricultural land ; Cropland ; Arable land ; Temporary crop ; Vegetables and melons ; Root, bulb or tuberous vegetables |
| Root, bulb or tuberous vegetable | Agricultural land ; Cropland ; Arable land ; Temporary crop ; Vegetables and melons ; Root, bulb or tuberous vegetables |
| Sapodilla tree | Agricultural land ; Cropland ; Permanent crops ; Fruit crop |
| Savannah | Natural vegetation ; Savannah |
| Savannah with shrubs | Natural vegetation ; Shrub land ; Savannah |
| Savannah with trees | Natural vegetation ; Open forest ; Savannah |
| Sesame | Agricultural land ; Cropland ; Arable land ; Temporary crop ; Oilseed crop |
| Shrub land | Natural vegetation ; Shrub land |
| Shrub vegetation | Natural vegetation ; Shrub land |
| Sorghum | Agricultural land ; Cropland ; Arable land ; Temporary crop ; Cereals |

| | |
|---|---|
| Soybean | Agricultural land ; Cropland ; Arable land ; Temporary crop ; Oilseed crop ; Leguminous |
| Sugarcane | Agricultural land ; Cropland ; Arable land ; Temporary crop  ; Sugar crop |
| Sunflower | Agricultural land ; Cropland ; Arable land ; Temporary crop ; Oilseed crop |
| Sweet potato | Agricultural land ; Cropland ; Arable land ; Temporary crop  ; Root/tuber crop with high starch or inulin content |
| Taro | Agricultural land ; Cropland ; Arable land ; Temporary crop ; Root/tuber crop with high starch or inulin content |
| Tea | Agricultural land ; Cropland ; Permanent crop ; Cash woody crop |
| Tomato | Agricultural land ; Cropland ; Arable land ; Temporary crop ; Vegetables and melons ; Fruit-bearing vegetables |
| Vegetable and root | Agricultural land ; Cropland ; Arable land ; Temporary crop ; Vegetables and melons ; Root, bulb or tuberous vegetables |
| Vegetables | Agricultural land ; Cropland ; Arable land ; Temporary crop ; Vegetables and melons |
| Vineyard | Agricultural land ; Cropland ; Permanent crop ; Cash woody crop |
| Water body | Water body |
| Watermelon | Agricultural land ; Cropland ; Arable land ; Temporary crop ; Vegetables and melons ; Fruit-bearing vegetables |
| Wattle tree | Agricultural land ; Cropland ; Permanent crop ; Multifunctionnal woody crop |
| Weakly vegetated agricultural | Agricultural land ; Cropland ; Arable land ; Temporary crop |
| Wetland | Natural vegetation |
| Wheat | Agricultural land ; Cropland ; Arable land ; Temporary crop ; Cereals |
| Wild radish | Agricultural land ; Cropland ; Arable land ; Temporary crop ; Cover crop |
| Woodlot | Agricultural land ; Cropland ; Permanent crop ; Multifunctionnal woody crop |
| Young fallow | Agricultural land ; Cropland ; Arable land ; Fallow |
| Zucchini | Agricultural land ; Cropland ; Arable land ; Temporary crop ; Vegetables and melons ; Fruit-bearing vegetables |

535