# Peer review of "Harmonized in situ datasets for agricultural land use mapping and monitoring in tropical countries"

_Earth System Science Data, 2021_

## Referee Comment (RC2)

This study has presented the ground data collected from nine selected sites in tropical countries within JECAM initiative. The area of site ranges from 250 to 11700 according to the location. This data is collected for either one or two years except Burkina Faso (7 years), Madagascar (4 years) and Brazil (3 years). As it is ground data collection manually – there is not accuracy assessment except set up of standardized procedures to collect data.

Overall, this work has potential to add value in the problem area of availability of ground data for agricultural monitoring and can be considered for publication with major edits. I have major and minor comments as follows:

Major comments

1. Why is the dataset called harmonized although it only depicts to be ground data?

2. What is the clear definition of classes used to label? Author may add these definitions for more clarity and avoid confusion? For example, what is mean by croplands in this study? (do croplands include agroforestry, rangelands and horticultural crops too)

3. For every site and region – summer crop and winter crop seasons are separate – if author is using this term – I would recommend to use it carefully as the seasons although name is same but months are different location-wise. In short , provide definition of winter crop and summer crop – and in the label – add unique name if possible

4. Figure 1 is totally misleading. The selected site cannot possibly represent the entire tropical farming systems. I would suggest to remove it as it is not adding any value. Main important is to highlight the sites and author can do that by showing site zoomed regions rather than misleading with unnecessary presentation.

5. Study site – the explanation of study sites in section 2.1 is repetitive of what's there is in table 1. I would recommend rewriting to avoid repetition.

6. Table 1 name is synthesis of database – which is not quite correct – I would recommend just naming it as "study area description" or related.

7. Table 1 can have additional columns such as season, temperature, major crops, average precipitation etc.

8. Data collection protocol need to provide with more details as it is important step in this data. I would recommend to explain it with sample examples of data points and showing the standardized format along with flowchart if possible.

9. Post processing of data may add many additional errors to the raw data point collection with the provided steps by author. I would suggest to provide more details and explanation about how the manual error were avoided? In short, provide the framework in methodological format. (although – the step is performed by same personnel who did survey – it is not valid explanation of expertise or scientific explanation)

10. In section data records – one of the column is data source – and there are three data sources – As it is important information – I would recommend author to provide a number on how many samples are "0" , "1" and "2" as data is mainly labelled as in situ

11. Detailed explanation on data source or data collection is needed regarding minimum size unit (MMU), labeling strategies, mix land use class and other details related to land use.

12. Total crop samples are ~20,257 and non-crop are related very low – how did you decide this number? Need further explanation on classes, their sampling size, sampling and labeling strategy.

13. What is the overarching goal and novelty in this dataset? I understand it is very important dataset – but author need to add its novelty and goal of research in introduction for more clarity.

Minor comments:

14. Line 40 and 41 – can be split to two sentences to avoid complexity

15. Validation through study cases – is confusing section – is it application of dataset or validation?

16. Overall, writing needs to improve for spelling and grammar – I would recommend professional English proof-reading – I had real difficult time in reading this paper.

17. Title of article contains "JECAM" – which needs to be expanded?

18. Abstract is misleading in many aspects such as – data time from 2013-2020 (which is not true as most of the sites has data from 2 years only). I would recommend author to be careful and precise facts in the abstract for more clarity and description of work.

19. For small field sizes – what was the strategy to collect data – how would it be homogeneous to the data collection strategy/

20. Overall, I would recommend author to improve readability of the article.

---

## Author Comment (AC1)

Comments from Referee 1:

Thank you for your positive comments and interesting suggestions. This document's intent is to provide point-by-point answers to your remarks, directly proposing, where possible, modifications to the original paper that will be integrated in the revised version. We have worked in particular on:

- mentioning similar efforts in northern countries with examples in Europe and USA
- describing in further details other available LULC reference data products
- providing more information about logistic details for field missions (field team, duration, partner training,…)
- putting forward future perspectives for the proposed database (BD update, non-JECAM data providers, …).

In remainder of the document, lines in **bold** echo your comments for ease of reading, lines in red provide direct answers to your comments, followed in case by proposed modifications to our paper (with new elements in green).

We sincerely hope that these corrections will match your expectations.
* * *
**The paper introduces a dataset of 27000 polygons in 7 countries (9 study areas) with information on land cover, crop type and cropping systems. This information can then be used to validate land use maps or train models for land use classification. It results from a large-scale international initiative supported by GEO and gathers scientists with a long experience in crop mapping with remote sensing data. It is also noteworthy that the 9 study areas represent very diverse agricultural areas.**

**The paper is clearly written and I advise for publication. My only minor revisions concern the following points:**

- **In the introduction: similar efforts in northern countries should be mentioned**

  ➔ In the introduction, we have added several paragraphs which quotes some similar efforts made in northern countries, along with an example on the way these data sets can be used for LC mapping (a reference has also been added).

Land use and land cover (LULC), and their changes, are key information to study and monitor carbon and water cycles, threats to biodiversity, but also to set up land use planning and public policies. In particular, accurate mapping of cropland and associated cropping practices is of primary importance for food security, agricultural and environmental monitoring as well as land management. However, cropland and crop type mapping using Earth observation data is still challenging as it requires large sets of training and validation data, and as the land use (field limits and content) generally changes annually, even seasonally. Large data sets on cropping practices are available in the Global North, mainly thanks to agricultural policies that support annual census and provide tools for the digitization at field level using Very High Resolution remote sensing imagery (e.g. the Land Parcel Identification System designed to implement common agricultural policy in the European Union, or the Cropland

Data Layer of the National Agricultural Statistic Services of the United States Department of Agriculture). Such data sets provide a very large number of annotated surface samples reporting yearly crop types, which can often easily be integrated in reference data sets for land cover mapping systems at the cost of a relatively simple "cleansing and harmonization" procedure (Inglada et al., 2017). Despite the fact that the declarative nature of such annotations makes them error-prone, such "noise" is typically compensated by the large number of available crop type samples. As arguable, no such large scale data base currently exists in most of the developing and emerging countries. Matter of facts, in these countries cropland and crop types can be particularly difficult to map (Waldner et al., 2015) because the fields are often small to medium size (Fritz et al., 2015), the crops are easily confused with natural vegetation and fallows, and cropping systems are typically highly variable in time and space. Each farming system has its own specificities in terms of crop type and composition, field size, cropping calendar, irrigated/rainfed mode and other practices (Bégué et al., 2018). It is thus necessary to adapt the classification approaches (satellite data and algorithms as well as training and validation in situ data) to the large variability of the farming systems in the world (Dixon et al., 2001), and thus to have access to appropriate training data.

- **In the methods: I think a little bit more details on the « logistics » issues would be interesting (how many people? how many days of field campaigns? etc). It is important that the reader understands how difficult it is to do field campaigns in southern countries.**

➔ In the Data collection section, we have added a paragraph with more detailed logistic information about the field team, duration, and partner training.

During a field mission, the team is composed by an agronomist with geoprocessing skills, accompanied by a national researcher or technician with expertise in the local farming systems and a local driver. In some countries (Burkina Faso, Senegal, Madagascar, Kenya), local partners were trained to collect data. The training sessions were carried out directly in situ to be the closer as possible of the reality. The data acquisition duration depends on the visited area: in Brazil (large fields and good road infrastructures), 300 plots can be visited in one day while for other sites (small to very small fields), it is possible to collect between 50 and 150 plots per day (depending on the roads state and fields accessibility). Usually, the mission for a smallholder's site of ~3600km² is one week with around 700 plots visited.

- **In the discussion/conclusion: add additional comments on future perspectives:**

➔ These are indeed very important remarks concerning the potential growth of the database in the future. We will provide a brief discussion on these points in the revised version.

  o **will the dataset will be updated regularly?**

First, recall that data collections have been performed in the framework of different projects and funded initiatives, which constitutes a significant part of our mission as well as of other research and

development institutions working on tropical agriculture. In this sense, this paper's aim is also to provide evidence that such independent field efforts can be mutualized to durably contribute to the extension of the database. To date, several field campaigns are already planned on some of the presented sites, and projects are being built which will lead to the inclusion of several new ones.

- o **how to add reference samples from non-JECAM colleagues?**

➜ The paper also proposes a set of technical guidelines for potential non-Jecam contributors, which might in turn lead to a significant extension of the geographic extent of the database and hence of its representativity with respect to the diversity of tropical agrosystems. Although no technical solution is proposed yet to facilitate external contributions, which will be pointed out as future work, we are definitely open to such contributions and willing to take care of the technical issues.

- o **how to improve the data collection in the future (UAV data, crowdsourcing)?**

➜ As we will mention in the revised introduction (see answer about details on existing land cover reference databases), we consider crowdsourcing a valuable approach for providing very large data sets, at the cost of a larger "noise" due to the loosely controlled quality of the contributions. Our field strategy is more meant to preserve quality at maximum, and provide a less extensive, yet reliable set of field observations. So we consider our approach as being somehow complementary with respect to crowdsourcing initiatives, and no such strategy is planned in the future.

UAVs (drones) can be, and already are in few of our sites, used to validate acquisitions and extend / accelerate the surveys over areas which are less accessible (far from roads, across fields / flooded areas, etc.). However, both the availability of UAVs as well as of the required competences to drive them is generally costly, so that, to date, it is very unlikely for such acquisition means to be systematically included in future campaigns. In some sites, the strong resemblance of some crop types when observed from and airborne sensors also prevents the use of such solutions.

Other comments are:

In the abstract,

**« Altogether, the datasets completed 27 074 polygons (20 257 crop and 6 817 non-crop) documented by detailed keywords. »**

**Depending on the authorized length of the abstract, it may be good to complete with additional information: how much maximum polygons/year and/or polygons/class and/or polygons/study area?**

➜ We have added some details about the minimum and the maximum plots visited in a year.

These quality-controlled datasets are distinguished by in situ data collected at field scale by local experts, with precise geographic coordinates, and following a common protocol. Altogether, the datasets completed 27 074 polygons (20 257 crop and 6 817 non-crop, ranging from 748 plots in 2013 (one site visited) to 5515 in 2015 (six sites visited) documented by detailed keywords. These datasets can be used to produce and validate agricultural land use maps in the tropics, but also, to assess the performances and the robustness of classification methods of cropland and crop types/practices in a large range of tropical farming systems. The dataset is available at https://doi.org/10.18167/DVN1/P7OLAP.

L61 to 70

**The list of reference datasets is interesting. I think it lacks some information on each of them (mainly, I am not sure they all discriminate various cropping systems or just croplands from other LC classes). In addition, I think it would be great to have this information in a table but that would imply to move it out from the introduction section to another section.**

➜ Indeed, more information about each product is interesting to understand the lack of agronomic data. We have finally chosen to provide further details on these products in the text (Introduction), since the different nature, approaches and objectives of the different data sets makes it difficult to make a rigorous summary through the use of a table.

At a global and continental scale, initiatives that freely distribute land cover reference datasets exist (see review by Tsendbazar et al. (2015)). The GOFC-GOLD (Global Observation for Forest and Land Cover Dynamics; see http://www.gofcgold.wur.nl/sites/gofcgold_refdataportal.php for further details and access to data) regroups and consolidates existing reference datasets used for the validation of legacy global land cover products (prior to 2015) at moderate spatial resolution (300m-1km) such as GLC 2000 and GlobCover 2005. All referenced databases are provided at global scale, ranging from few hundreds to around 2,000 samples each. Except for GlobCover 2005, which contains a "rainfed cropland" class, other referred LC nomenclatures only contain a single cropland class, sometimes referred to as "cultivated".

Other data collection experiences reached a sensibly higher number of samples through the use of crowdsourcing campaigns, a notable example being the LULC reference dataset presented in Fritz et al., 2017, and its companion work from Laso Bayas et al., 2017b: thanks to the Geo-Wiki tool providing an easy-to-use interface for the photo-interpretation of very high spatial resolution satellite images, it was possible to collect up to 150,000 samples of different LULC classes. This includes over 36,000 cropland locations, distributed over contrasted areas in terms of cropland density. As in the previous case, a single cropland class is referenced in the nomenclature, alone or mixed with natural vegetation ("mosaic" class). Although crowdsourcing confirms as a valuable strategy to collect reference cropland data at larger scales, it still remains unsuited when precise information has to be collected, both spatially (resolution, plot boundaries, etc.) and in terms of crop type nomenclatures. Matter of facts, most of the crowdsourcing initiatives are based on visual image interpretation, which prevents the precise localization and identification of cropping practices. Shifting to a crowdsourced field strategy

will not be suitable as well, both because of the very specific agronomic and GIS competences needed and the limited accessibility to cultivated areas in tropical countries.

More lately, the LandCoverNet dataset has been released for the African continent (Alemohammad et al., 2020), with the specific aim to foster the use of recent machine and deep learning approaches for automatic land cover classification. Here, samples are provided in the form of densely annotated image chips (256x256 pixels at 20m resolution) accompanied by the corresponding Sentinel-2 observations over the reference year (2018). A total number of 1,980 fully annotated chips, accounting for more than 30 million of labelled pixels, are provided, spanning 66 tiles of Sentinel-2 over the entire African continent. Although such dataset could allow a finer spatial validation of LULC products at high resolution, it still provides a single "cultivated land" class, making it unsuitable for the assessment of LULC products specifically conceived for the monitoring of agricultural systems.

**In addition, it would be interesting to get more info on such initiatives in northern countries (in Europe and in the US mainly)**

➔ See answer above

**Figure 1. It looks like there is no croplands in Europe, US and Australia. Maybe you could add a word on that in the figure caption, to explain it only focuses on developing/emerging countries**

➔ The caption has been completed as suggested

[Figure]

Figure 1. Location map of the study sites, and the associated number of collection years and sampled plots (symbolized by the size of the red circles), displayed on the FAO (broad) farming system map focused on developing / emerging countries (Dixon et al., 2001)

**L.184. « Field surveys were conducted yearly ». Yearly is not very well chosen since you have only one year of data for a few sites.**

➔ Indeed, it was a shortcut and we have corrected the sentence.

The acquisition protocol is based on the JECAM guidelines (Defourny et al., 2014) with adaptations to consider some characteristics of tropical agriculture (mainly small field size and accessibility). Field surveys were conducted at least once in each study zone, with several sites revisited over multiple consecutive years (up to 7 for the Burkina Faso site). Campaigns took place either around the growing peak of the cropping season, for the sites with a main growing season linked to the rainy season such as Burkina Faso, or seasonally, for the sites with multiple cropping (e.g. São Paulo site). Except for Senegal where a stratified sampling plan for field surveys was used (Ndao et al., 2021), the GPS waypoints were gathered following an opportunistic sampling approach (called the "windshield survey") along the roads or tracks according to their accessibility (that can be difficult during the rainy season, leading to less surveys in secondary roads or tracks in some study areas), while ensuring the best representativity of the existing cropping systems in place (Defourny et al., 2014; Waldner et al., 2019).

**Section 2.2. I would be interested in reading more information on the number of colleagues who participated to the data collection and if it was necessary to train local colleagues to collect the data. (This may also appear later, around L 250). The capacity building part is important to ensure future update of the database by local partners.**

➔ See answer above

**In Table 2, a fourth column with an example for a given polygon of the database would be welcome**

➔ We have added a 4th column with an example

| Attribute Name | Data Type | Description / available arguments | Example |
|---|---|---|---|
| Id | Numeric | Unique ID | *26413* |
| Country | Text | Country name | *Burkina Faso* |
| SiteName | Text | Site name (generally related to the biggest city around or to the region name) | *Koumbia* |
| DataSource | Numeric | Discrimination between land uses acquired from *in situ* surveys or satellite image CAPI (computer assisted photointerpretation)
0: Land use from *in situ* survey
1: Land use from satellite image interpretation
2: Land use from aircraft observation | *0* |
| AcquiDate* | Date | *In situ* survey acquisition date or satellite image acquisition date (when the land use is photointerpreted, see "DataSource" attribute) – Format: yyyy-mm-dd | *2020-10-21* |
| LandCover | Text | Land cover of the polygon. If value is "Cropland", see CropType 1, 2 and 3 attributes for more information | *Cropland* |
| CropType1 | Text | Main crop type of the polygon | *Cotton* |

| | | | |
|---|---|---|---|
| CropType2 | Text | Secondary crop type of the polygon (in case of intercropping) | *Maize* |
| CropType3 | Text | Tertiary crop type of the polygon (in case of intercropping) | *NULL* |
| SOS* | Date | Start of season date in the site (if empty, this means that no specific season exists in the study area) – Format: yyyy-mm-dd | *2020-05-01* |
| EOS* | Date | End of season date in the site (if empty, this means that no specific season exists in the study area) – Format: yyyy-mm-dd | *2020-11-30* |
| Irrigated | Numeric | Presence/absence of an irrigation system
   0: No information available
   1: Rainfed
   2: Irrigated
   Empty: For polygons other than cropland | *1* |
| Intercrop | Numeric | Presence/absence of intercropping
   0: Single crop
   1: Mixed crop or row inter-crop
   2: Agroforestry
   Empty: For polygons other than cropland | *1* |
| Weeding | Numeric | Presence/absence of weeds
   0: No information available
   1: Presence of weeds
   Empty: For polygons other than cropland | *0* |
| Area_ha | Numeric | Polygon area in hectares | *0.446* |
| KeyWords | Text | Set of terms associated to the land use of the polygon (separated by semicolons ";") | *Agricultural land ; Cropland ; Arable land ; Temporary crop ; Cash crop ; Fiber crop* |

**In figure 3, I guess the pasture class mentioned in the description of some study areas are included in the grassland class. Yet I wonder if there is a discrimination between natural grasslands and managed pastures. can you please clarify that point?**

➔ Indeed, Pasture should be overally considered as a particular type of Grassland cover whose vegetation dynamic may be affected by the regular passage of livestock, and should not be confused with cultivated pastures for which the database eventually reports the specific crop type. In order to preserve at maximum the information about the possible use of grassland for animal feeding, our global strategy across the different sites is to note grassland areas as pastures whenever such use can be assumed with certainty, either from prior knowledge on the visited areas or by the direct observation of the presence of livestock.

**L255 « Finally, the fact that the same person performed the whole acquisition and processing chain - from waypoint collection to polygon labelling - minimizes errors and contributes to the overall quality of the datasets. »**

**This point is questionabale. If the operator is not « good », he may repeat the same error N times. More generally, operators working with photointerpetation usually work with a cross-checking protocol to minimize errors. But I think that in your case it is a bit different since the class labelling is done by at least two people (L249) while this step mentioned in that sentence (L255) only regards the geoprocessing part (polygon delineation).**

**I think you should rephrase slightly to clarify this point.**

➔ Again, this phrase resumes too briefly the concept of per-site field supervision, which is actually set up in a more structured way in order to ensure a reliable quality checking. In order to clarify this point, we have added a short paragraph in the Quality checking section, to explain that each site has a referee person who supervises the whole field collection chain, whose different steps, from the acquisition to the integration in the database, are generally performed by a dedicated specialist.

4.1 Quality Checking

Finally, each site has a referee person who knows very well the area. He supervises all the chain from the data collection to the database integration. Following this approach, each step is generally conducted by one or more "discipline" specialists (agronomy, GIS, database) whose work is coordinated by the referee in order to minimize errors and contribute to the overall quality of the data sets.

**For the other comments:**
**L103 . « 60 x 60 km² area ». I am not English but I would say « 60 x 60 km area » (thus removing the ²) : 60 x 60 km = 3600 km². Please have a check.**
**L103. « commune ». sounds very French. Maybe put in italics ?**
**L154 : March instead of Mars**
**L197 « filling » instead of « filing »**
**L254 . « photographs » instead of « photos »**
**L270 « In Table 3, are given the type and extent of the zones where are located our JECAM study sites, for both maps.»**
**I would rephrase as follos : « For both maps, the type and extent of the zones corresponding to our JECAM study sites are given in Table 3 »**
**Table 3. There is a double parenthesis in line 1.**
**L289 First, « , » is missing**
**L303 « valorized » rephrase « used » ?**
**L303 Tocantins . Do not separate the S.**
**L314. I guess the citation should be Jolivot et al. (2021)**
**L321 JECAM (not JEAM)**

➔ All these remarks will be corrected in the revised text. Thanks very much again for your valuable implication and comments.

---

## Author Comment (AC2)

We have thoroughly revised our manuscript according to the comments and suggestions provided by the reviewer. We would like to thank her/him for its review, which allowed us to improve our manuscript.  All the new and modified contents are in blue in the revised version of the manuscript.

In remainder of the document, lines in **bold** echo your comments for ease of reading, lines in red provide direct answers to your comments, followed in case by proposed modifications to our paper (with new elements in green).

We sincerely hope that these corrections will match your expectations.
* * *
**This study has presented the ground data collected from nine selected sites in tropical countries within JECAM initiative. The area of site ranges from 250 to 11700 according to the location. This data is collected for either one or two years except Burkina Faso (7 years), Madagascar (4 years) and Brazil (3 years). As it is ground data collection manually – there is not accuracy assessment except set up of standardized procedures to collect data. Overall, this work has potential to add value in the problem area of availability of ground data for agricultural monitoring and can be considered for publication with major edits.**
**I have major and minor comments as follows:**

**Major comments**
**1. Why is the dataset called harmonized although it only depicts to be ground data?**

➔ In fact it is the ground data sets collected on each of the nine sites that are harmonized with each other, from data collection to final database format ready to use. We prefer the term "harmonized" to "standardized", because part of the work was done *a posteriori*.

**2. What is the clear definition of classes used to label? Author may add these definitions for more clarity and avoid confusion? For example, what is mean by croplands in this study? (do croplands include agroforestry, rangelands and horticultural crops too)**

➔ In the "2.2 Data Collection" section the attribute "keywords" is described (see extract below) and aims to provide harmonized and generic keywords describing each class. These keywords are based on existing standards in terms of land use definitions (FAO, 2020 - see link below) that can be consulted to have a clear definition of each class or term used. When the JECAM nomenclature showed disagreements with the FAO land use definitions (eg. In the FAO definitions, the "fallows" class is considered as cropland, while it is not in JECAM nomenclature), priority was given to the FAO definitions. The full list of the land use classes and their associated keywords are now provided in Appendix B.

Extract of "2.2 Data Collection" Section:

*"An attribute referred to as "Keywords" was also created in order to associate various generic terms (land cover, crop group, crop type, cropping practice, etc. (Appendix B)) to each polygon. This attribute has two objectives: (i) facilitating keyword search for the user, (ii) allowing the user to create his own nomenclature (hierarchic or not) with different levels of detail so that the nomenclature can be dedicated to the user's needs. These terms are based on the FAO land use definitions (FAO, 2020) and JECAM hierarchic nomenclature (Defourny et al., 2014), which were adapted to take into account the diversity of the farming systems in the surveyed sites."*

Link of FAO land use definitions (see "References"):
FAO: Land use, irrigation and agricultural   USE, IRRIGATION AND AGRICULTURAL PRACTICES - DEFINITIONS:
http://www.fao.org/fileadmin/templates/ess/ess_test_folder/Definitions/Land_Use_Definitions_FAOSTAT.xlsx, last access: 9 September 2020.

**3. For every site and region – summer crop and winter crop seasons are separate – if author is using this term – I would recommend to use it carefully as the seasons although name is same but months are different location-wise. In short , provide definition of winter crop and summer crop – and in the label – add unique name if possible.**

➔ We agree with the reviewer, and it is why in the database we have the attributes Start of Season (SOS) and End of Season (EOS) that gives the validity period for the crop type recorded for each polygon (see Table 1), and why the terms "winter crop" and "summer crops" do not appear in the keywords list (see Appendix B).
However, we maintained the term "summer" and "winter crops" in the description of the Brazilian sites (Section 2.1) because the months concerned are specified, but in Table 1, to avoid confusion, we replaced "For each field in the Tocantins site, the operator recorded the crop type of the 2 seasons (summer / winter) by observing the crop residues on the field or by interviewing the farmers" by
*"For each field at the Tocantins site, the operator was able to record the crop type for the two cropping seasons by observing the crop residues in the field or by interviewing the farmers".*

**4. Figure 1 is totally misleading. The selected site cannot possibly represent the entire tropical farming systems. I would suggest to remove it as it is not adding any value. Main important is to highlight the sites and author can do that by showing site zoomed regions rather than misleading with unnecessary presentation.**

➔ We regret that the representation in Figure 1 may have given the impression that we consider our sites to be representative of the entire tropical farming systems, while we just intended to illustrate the diversity of our sites in terms of tropical agrosystems. But as Figure 1 is misleading, we have simplified it to provide only information on the location of the sites.

[Figure]

*Figure 1. Location map of the study sites, and the associated number of collection years and sampled plots (symbolized by the size of the red circles).*

**5. Study site – the explanation of study sites in section 2.1 is repetitive of what's there is in table 1. I would recommend rewriting to avoid repetition.**

➜ Thank you for bringing this to our attention. There is probably a misunderstanding about Table 1 (now Table 2 in the new version of the manuscript) due to its positioning prior to the database description. Indeed, the table mainly provides statistics about the final database, such as the total number of records per-site (polygons), their average size, etc. That is why we named it "Database synthesis".
To avoid confusion, Table 1 was moved to section 3 Data Records, just after the database description and corresponds now to Table 2; the column "cropping pattern" and the size of the study site were removed because of information redondance with the text in section 2.1.

**6. Table 1 name is synthesis of database – which is not quite correct – I would recommend just naming it as "study area description" or related.**

➜ Please, cf. to the answer to Comment 5. We hope that in the revised version of the paper the description of the sites and the description of the database will be clearer.

**7. Table 1 can have additional columns such as season, temperature, major crops, average precipitation etc.**

➜ As explained in the answer to Comment 5, Table 1 was not intended to report geographical description of the sites, which is indeed provided in the text of Section 2.1. Most of the requested information is already in the aforementioned text. Sorry again for the misunderstanding.

**8. Data collection protocol need to provide with more details as it is important step in this data. I would recommend to explain it with sample examples of data points and showing the standardized format along with flowchart if possible.**

➜ Data collection protocol is indeed a very important step of our work. In order to help the understanding, we added a workflow of the data acquisition in the Section 2.2, showing: the field data

entry form filled for one plot, GPS waypoints and then boundaries digitized for each GPS point, displayed on a satellite image in false color.

[Figure]

*Figure 3. Workflow of the data acquisition: (a) field data entry form used on the GPS tablet, (b) GPS waypoints acquired in the field and (c) corresponding plots after digitalization of the boundaries, displayed on a satellite image in false color (Red: Near Infrared band, Green: Red band, Blue: Green band).*

**9. Post processing of data may add many additional errors to the raw data point collection with the provided steps by author. I would suggest to provide more details and explanation about how the manual error were avoided? In short, provide the framework in methodological format. (although – the step is performed by same personnel who did survey – it is not valid explanation of expertise or scientific explanation)**

➜ Digitization of each field or non-crop entity boundaries was performed on very high spatial resolution images that were acquired specifically for this purpose just before the field campaigns. This step was done quickly after the field campaigns and in the same order as the GPS waypoints records so that the operator can well remember each one. This digitization was performed by remote sensing experts who also participated in the field surveys, and had very good skills in visual interpretation of satellite images. Furthermore, photographs were available for each waypoint to ensure the consistency between the ground and satellite information. Finally, no doubts on entities boundaries were accepted, and the entities having fuzzy boundaries or too much heterogeneity were discarded.

**10. In section data records – one of the column is data source – and there are three data sources – As it is important information – I would recommend author to provide a number on how many samples are "0" , "1" and "2" as data is mainly labelled as in situ**

➜ To illustrate the relative importance of the data source, the following figure was added in the document :

[Figure]

*Figure 6. Distribution of the data sources, given in percentage of the total number of polygons per site.*

**11. Detailed explanation on data source or data collection is needed regarding minimum size unit (MMU), labeling strategies, mix land use class and other details related to land use.**

➔ In section 2.2, we refer to a minimum homogeneous field size which can be considered for waypoints recording of 0.04 ha. We modified the text to explicitly mention the notion of Minimum Sampling Unit.

"Waypoints were only recorded for homogenous fields/entities of at least 20 x 20 m² (against a *minimum sampling unit* of 0.25 ha with a minimum width of 30 m in JECAM guidelines)."

Regarding the other information acquisition (such as labeling strategies, mix land use class, irrigation,..), we used a data entry form to facilitate the data entry : scrollable lists, checked box,... Some examples of lists are now displayed on Appendix A.

**12. Total crop samples are ~20,257 and non-crop are related very low – how did you decide this number? Need further explanation on classes, their sampling size, sampling and labeling strategy.**

➔ This is an important point. Overall speaking, the number of crop vs. non-crop samples is not fixed beforehand in each field campaign, but is a direct consequence of the opportunistic sampling approach described in Section 2.2. In other words, these numbers reflect the ratio of crop vs. non-crop surfaces observed along the covered tracks. However, literature on remote sensing based crop mapping shows that the problem of "extracting cropland" in a landscape is a relatively easier task with respect to detecting different agricultural land uses (e.g. crop types). This also means that the need for annotated surfaces over crop classes is definitely more important that for non-crop ones. Of course, this does not mean that the proposed database can be considered exhaustive for non-crop classes, but this is not its purpose. Concerning sampling strategies, there is no difference between crop and non-crop surfaces.

**13. What is the overarching goal and novelty in this dataset? I understand it is very important**

**dataset – but author need to add its novelty and goal of research in introduction for more clarity.**

➔ Yes, you are right. We need to emphasize the dataset quality and the final goal. We added a part at the end of the introduction section in that respect:

*The experiment has been operating since 2013, and some in situ datasets produced at the field scale have been used in different benchmarking mapping studies (Waldner et al., 2016; Inglada et al., 2015). However, only a part of the collected ground data was used in these studies and the databases are not publicly shared.*

*To make agricultural land use data publicly available to the remote sensing community, for classification algorithm benchmarking or LULC product validation for example, an important work of harmonization of in situ JECAM and JECAM-like agricultural land use datasets was undertaken for nine sites located in the tropical belt. The acquisition protocol was adapted from Defourny et al. (2014) to take into account the characteristics of tropical agriculture (e.g. small field size, accessibility). At each site, information on crop type and cropping practices was collected locally, at the field level, with a detailed nomenclature. The acquisition period was between 2013 and 2020, and the number of monitoring years per site was between 1 and 7.*

*In this paper, we describe in detail the study sites, the data collection protocol and the structure of the final database. We then discuss how the harmonization of the dataset and the diversity of the studied agrosystems, including small-holder farming, make our dataset unique and valuable for applications in the emerging/developing countries in the tropics.*

**Minor comments:**
**14. Line 40 and 41 – can be split to two sentences to avoid complexity**

➔ Indeed, the sentence is very long and not very pleasant to read. We splitted it in 2 sentences as recommended.

*These datasets can be used to produce and validate agricultural land use maps in the tropics. They can also be used to assess the performances and robustness of classification methods of cropland and crop types/practices in a large range of tropical farming systems.*

**15. Validation through study cases – is confusing section – is it application of dataset or validation?**

➔ Thank you for this comment. The sub-section "4.3. Validation through study cases" was removed from the "4. Technical validation" section, and converted into a new section titled to "5. Dataset application study cases".

**16. Overall, writing needs to improve for spelling and grammar – I would recommend professional English proof-reading – I had real difficult time in reading this paper.**

➔ We are sorry to hear that you had difficult time reading the paper. The revised version has been reviewed by a professional English native speaker (cf. the join certificate).

**17. Title of article contains "JECAM" – which needs to be expanded?**

➔ In order not to make the title of the article too long, the word JECAM has been removed from the title. However, the word "JECAM" appears in the expanded form in the abstract.

**18. Abstract is misleading in many aspects such as – data time from 2013-2020 (which is not true as most of the sites has data from 2 years only). I would recommend author to be careful and precise facts in the abstract for more clarity and description of work.**

➔ Thank you for this comment. In order to be more precise, we changed the text from *"In this paper, we present nine datasets collected in a standardized manner between 2013 and 2020 in seven tropical and subtropical countries within the framework of the international JECAM (Joint Experiment for Crop Assessment and Monitoring) initiative"*,

to

"In this paper, we present a database made of 24 datasets collected in a standardized manner over nine sites within the framework of the international JECAM (Joint Experiment for Crop Assessment and Monitoring) initiative; the sites were spread over seven countries of the tropical belt, and the number of data collection years depended on the site (from 1 to 7 years between 2013 and 2020)".

**19. For small field sizes – what was the strategy to collect data – how would it be homogeneous to the data collection strategy/**

➔ No specific strategy was applied for small field sizes. On the protocol, the minimum sampling size was 20*20m (0.04 ha) and attention focused primarily on the homogeneity of the fields, both on the ground and on satellite images used for boundary delineation.

**20. Overall, I would recommend author to improve readability of the article.**

➔ We hope that the English proof-reading and the revisions following the reviewer's comments have improved the readability of the article.

We want to thank you again for your thoughtful reading.

---

## Author Comment (AC3)

**Editing Certificate**

This document certifies that the manuscript

**Harmonized in situ datasets for agricultural land use mapping and monitoring in tropical countries**

prepared by the authors

**Audrey Jolivot, Valentine Lebourgeois, Louise Leroux, Raffaele Gaetano, Agnes Begue**

was edited for proper English language, grammar, punctuation, spelling, and overall style by one or more of the highly qualified native English speaking editors at AJE.

This certificate was issued on **November 3, 2021** and may be verified on the AJE website using the verification code **B958-7AE9-ADD5-06BC-BD33** .

[Figure]

Neither the research content nor the authors' intentions were altered in any way during the editing process. Documents receiving this certification should be English-ready for publication; however, the author has the ability to accept or reject our suggestions and changes. To verify the final AJE edited version, please visit our verification page at aje.com/certificate. If you have any questions or concerns about this edited document, please contact AJE at support@aje.com.

AJE provides a range of editing, translation, and manuscript services for researchers and publishers around the world.
For more information about our company, services, and partner discounts, please visit aje.com.